# Co-differential Gene Selection and Clustering Based on Graph Regularized Multi-View NMF in Cancer Genomic Data

**DOI:** 10.3390/genes9120586

**Published:** 2018-11-28

**Authors:** Na Yu, Ying-Lian Gao, Jin-Xing Liu, Junliang Shang, Rong Zhu, Ling-Yun Dai

**Affiliations:** 1School of Information Science and Engineering, Qufu Normal University, Rizhao 276826, China; yunacsw@126.com (N.Y.); shangjunliang110@163.com (J.S.); zhurongsd@126.com (R.Z.); dailingyun_1@163.com (L.-Y.D.); 2Library of Qufu Normal University, Qufu Normal University, Rizhao 276826, China; yinliangao@126.com

**Keywords:** non-negative matrix factorization, integrated model, graph regularization, common differential gene selection, multi-view clustering

## Abstract

Cancer genomic data contain views from different sources that provide complementary information about genetic activity. This provides a new way for cancer research. Feature selection and multi-view clustering are hot topics in bioinformatics, and they can make full use of complementary information to improve the effect. In this paper, a novel integrated model called Multi-view Non-negative Matrix Factorization (MvNMF) is proposed for the selection of common differential genes (co-differential genes) and multi-view clustering. In order to encode the geometric information in the multi-view genomic data, graph regularized MvNMF (GMvNMF) is further proposed by applying the graph regularization constraint in the objective function. GMvNMF can not only obtain the potential shared feature structure and shared cluster group structure, but also capture the manifold structure of multi-view data. The validity of the proposed GMvNMF method was tested in four multi-view genomic data. Experimental results showed that the GMvNMF method has better performance than other representative methods.

## 1. Introduction

With the rapid development of gene sequencing technology, a large number of multi-view data have been generated. In essence, multi-view data are insightful and have multiple levels of genetic activity information. Exploring this information will provide us with an unprecedented opportunity to discover the molecular mechanisms of cancer [1]. The Cancer Genome Atlas (TCGA) is the largest genome-based platform. And it provides a large number of different types of omics data. In this paper, we use gene expression (GE), copy number variation (CNV), and methylation (ME) data of four cancers in the TCGA database. They are mutually dependent on each other [2].

In the field of bioinformatics, feature selection and clustering are two important ways to explore genomic data [3,4]. To some extent, feature selection can reduce computational complexity and also find differentially expressed genes associated with cancer. It promotes cancer research at the molecular level of genes. Multi-view clustering is the division of samples or genes in a multi-view dataset into several subsets based on their potential group structure; the samples or genes in the same subset have similarities.

With the advent of the big data era, data are no longer single-view but multi-view data composed of different sources. The information of multiple views in multi-view data is complementary, and it is of great significance to conduct in-depth research on this information [5]. Therefore, a multi-view model that specializes in multi-view data has emerged. By learning a multi-view model, we can mine multiple views to achieve better performance than relying on a single view.

Since non-negative matrix factorization (NMF) is an effective matrix decomposition method, more and more researchers are paying attention to the integration model of NMF [6,7]. The integrated model of NMF has straightforward interpretability. Naturally, some improvements and enhancements to the integration model of NMF have been proposed. For example, Zhang et al. [2] proposed joint NMF (jNMF) to discover the common pattern in multi-dimensional genomic data. In order to enhance the robustness of the model to heterogeneous noise, Yang et al. [8] obtained the integrative NMF (iNMF) model by improving jNMF. After applying orthogonal constraints on jNMF, Stražar et al. [9] proposed integrative orthogonality-regularized NMF (iONMF) to predict protein-RNA interactions. In order to detect differentially expressed genes in transcriptomics data, Wang et al. [10] proposed a new method called joint non-negative matrix factorization meta-analysis (jNMFMA) by combining jNMF and meta-analysis.

Although the above methods are effective, their flexibility is limited. They cannot simultaneously perform feature selection and multi-view clustering. However, feature selection and multi-view clustering facilitate the multi-level understanding of the overall system of cancer. Therefore, in this paper, we present a novel method called Multi-view Non-negative Matrix Factorization (MvNMF). It can effectively perform selection of co-differential genes and multi-view clustering simultaneously. Specifically, we improved jNMF by decomposing the coefficient matrix $H_{I}$ into the product of the subspace transformation matrix $U_{I}$ and the shared coefficient matrix V. Then, the new model is $X_{I}=WU_{I}V$
(I=1,2,…,d). d indicates the number of different types of non-negative matrices. The shared basis matrix W contains potential group structures between different views. The shared coefficient matrix V not only has the low rank characteristic, but also has the shared feature pattern for views with different sources. In order to enhance the robustness of MvNMF to data containing manifold structures, the graph regularized MvNMF (GMvNMF) method is further proposed. This can be obtained by adding the intrinsic geometric information of the data to the MvNMF method.

The main contributions of this paper are as follows:In order to effectively cluster and select features for multi-view data at the same time, a novel integrated model called MvNMF is proposed. In the MvNMF framework, the shared basis matrix can reconstruct the potential cluster group structure, which contributed to the improvement of clustering performance. The selection of the co-differential genes can be performed because the shared coefficient matrix can recover the common feature pattern from different views.The graph regularization was applied to the objective function to form the GMvNMF method, which ensured that GMvNMF can capture the manifold structure of the multi-view data. This makes sense for the performance improvement of the integrated model.Scientific and rational experiments were designed on the cancer genomic data to illustrate the validity of the GMvNMF method and achieve satisfactory results.

In what follows, jNMF and its representative variants, graph regularization are reviewed in Section 2; a detailed description of the proposed GMvNMF method is also included. The results of multi-view clustering and co-differential gene selection are presented in Section 3. Finally, the paper is concluded in Section 4.

## 2. Materials and Methods

### 2.1. Joint Non-Negative Matrix Factorization and Representative Variants

Joint NMF (jNMF) [2] is a popular matrix decomposition algorithm. In the field of bioinformatics, each type of genomic data can be represented as an original matrix. The row of the matrix represents the sample, and the column represents the expression level of genomic data.

Given d different types of non-negative matrices, the goal of jNMF is to find that the product of a shared basis matrix $W_{n\times k}$ and the corresponding coefficient matrix ${\left(H_{I}\right)}_{k\times m_{I}}$ is similar to the input data matrix ${\left(X_{I}\right)}_{n\times m_{I}}$, i.e., $X_{I}\approx WH_{I}(I=1,2,\ldots ,d)$. n represents the number of rows of the input data matrix. The value of k means the degree of dimensionality reduction of data. $m_{I}$ represents the number of columns of the i-th input data matrix. Further, the objective function of jNMF can be expressed as:(1)$$\min\sum_{I=1}^{d}{\left\|X_{I}-WH_{I}\right\|}_{F}^{2}\;s.t.\;W\geq 0,H_{I}\geq 0,$$
where ${\left\|\cdot \right\|}_{F}$ denotes the Frobenius norm of the matrix. The shared basis matrix can reflect the sharing pattern of multi-view data matrices from different sources [11]. It is obvious that jNMF is equivalent to NMF when the value of d is 1. In other words, jNMF is a flexible and clever NMF extension model for the integration of multi-view data. Then, the updated rules are as follows:(2)$$W_{il}\leftarrow W_{il}\frac{\sum_{I=1}^{d}{\left(X_{I}H_{I}\right)}_{il}}{\sum_{I=1}^{d}{\left(WH_{I}H_{I}^{T}\right)}_{il}},$$
(3)$$H_{lj}\leftarrow H_{lj}\frac{{\left(W^{T}X_{I}\right)}_{lj}}{{\left(W^{T}WH_{I}\right)}_{lj}},\;I=1,2,\ldots ,d,$$
where $W_{il}$ and $H_{lj}$ refer to specific elements in the matrices W and $H_{I}$.

jNMF is effective in finding homogenous effects of data from different sources, however, it does not consider the effects of heterogeneous noise between multi-view data. Therefore, Yang et al. [8] proposed an integrative NMF (iNMF) model. Specifically, the objective function of iNMF was defined as follows:(4)$$\begin{array}{l} \min\sum_{I=1}^{d}{\left\|X_{I}-\left(W+V_{I}\right)H_{I}\right\|}_{F}^{2}+\lambda \sum_{I=1}^{d}{\left\|V_{I}H_{I}\right\|}_{F}^{2}\\ s.t.\;W\geq 0,H_{I}\geq 0,V_{I}\geq 0,I=1,\ldots ,d, \end{array}$$
where λ represents a balance parameter. $WH_{I}$ means taking into account the homogenous effect, and $V_{I}H_{I}$ means considering the heterogeneous effect. In other words, $V_{I}H_{I}$ can be used as an approximation of the heterogeneous effect.

In order to obtain non-overlapping and sparse solutions, Stražar et al. [9] proposed integrative orthogonality-regularized non-negative matrix factorization (iONMF) by applying orthogonal regularization in the jNMF framework. Thus, its cost function can be written as:(5)$$\begin{array}{l} \min\sum_{I=1}^{d}\left({\left\|X_{I}-WH_{I}\right\|}_{F}^{2}+\alpha {\left\|H_{I}H_{I}^{T}-I\right\|}_{F}^{2}\right)\\ s.t.\;W\geq 0,H_{I}\geq 0,I=1,\ldots ,d, \end{array}$$
where I represents a unit matrix, and α is a trade-off parameter. iONMF exhibits better performance in predicting protein-RNA interactions.

### 2.2. Graph Regularization

The maturity of manifold learning theory made people pay more attention to the internal geometric structure in the original data. The basic idea of graph regularization is to reconstruct the low-dimensional manifold structure embedded in high dimensional sample space. That is to say, adjacent sample points in the high dimensional sample space should be as close as possible in the low dimensional space [12]. If each sample point is used as the vertices of the graph to construct a K-nearest neighbor graph, then a symmetric matrix E is obtained [13]. $E_{ij}$ represents the weight of the edge connecting vertex i and vertex j. Thus, the degree of proximity between the vertices can be measured using $E_{ij}$. The definition of $E_{ij}$ can be shown as follows:(6)$$E_{ij}=\{\begin{array}{l} 1\;\mathrm{if}\;x_{i}\in N_{K}\left(x_{j}\right)\;\;\mathrm{or}\;x_{j}\in N_{K}\left(x_{i}\right),\\ 0\;\mathrm{otherwise}, \end{array}$$
where $N_{K}\left(x_{i}\right)$ is the set of K nearest neighbors of $x_{i}$. $x_{i}$ is the i-th sample point.

Mathematically, the graph regularization can be formulated as follows:(7)$$\begin{array}{l} \;\underset{V}{\min}\sum_{i,j}{\left\|s_{i}-s_{j}\right\|}^{2}E_{i,j}\\ =\underset{V}{\min}\mathrm{tr}\left(V\left(D-E\right)V^{T}\right)\\ =\underset{V}{\min}\mathrm{tr}\left(VLV^{T}\right), \end{array}$$
where D is a diagonal matrix and the elements on the diagonal are composed of the sum of the rows or columns of E. $s_{i}$ and $s_{j}$ are the low dimensional representation of $x_{i}$ and $x_{j}$, respectively. tr(⋅) is the trace of the matrix. The matrix V represents the coefficient matrix produced by the decomposition of NMF. Finally, L=D−E is a graph Laplacian matrix [14].

### 2.3. Graph Regularized Multi-View Non-Negative Matrix Factorization

It is well known that cancer genomic datasets contain many types of data. In order to effectively utilize the information of multiple views, we proposed the MvNMF model and further improved it to get GMvNMF. The GMvNMF algorithm is introduced in detail; the proposed algorithm is given in Algorithm 1.

#### 2.3.1. Objective Function

jNMF is a good integration model that can fully explore potential shared structures in multiple views [2]. However, its flexibility is not sufficient to explore multi-view clustering and selecting co-differentially expressed genes at the same time. For multi-view clustering, the same sample points in different views are likely to be grouped together. Therefore, we required the basis matrix to exhibit the potential cluster structure that is shared by different views. For the selection of co-differential genes, the expression of the same gene in different views should be considered comprehensively. Therefore, we required that the coefficient matrix reflected the shared feature structure from different views.

In view of the above requirements, a model called MvNMF was designed. It can simultaneously perform multi-view clustering and selection of co-differentially expressed genes. ${\left(X_{I}\right)}_{n\times m_{I}}$ can be approximated by $WU_{I}V$
(I=1,2,…,d). Specifically, the objective function of MvNMF can be formulated as optimization problem:(8)$$\begin{array}{l} \min\sum_{I=1}^{d}{\left\|X_{I}-WU_{I}V\right\|}_{F}^{2}\;\\ \;s.t.\;W\geq 0,U_{I}\geq 0,V\geq 0,\;I=1,2,\ldots ,d, \end{array}$$
where $W_{n\times k}$ is the shared basis matrix, ${\left(U_{I}\right)}_{k\times r}$ is the subspace transformation matrix, and $V_{r\times m}$ is the shared coefficient matrix. n is the number of samples in the dataset. m is the number of features in the dataset. k denotes dimensionality reduction and r denotes the rank of the matrix. From the objective function we can see that the $H_{I}$ of jNMF can be approximated by $U_{I}V$.

We further considered the low-dimensional manifold structure embedded in the high-dimensional multi-view data space. Thus, the GMvNMF model was obtained by combining graph regularization and MvNMF.

Its objective function can be written as the following minimization problem:(9)$$\begin{array}{l} \min\sum_{I=1}^{d}{\left\|X_{I}-WU_{I}V\right\|}_{F}^{2}\;+\lambda_{I}\mathrm{tr}\left(W^{T}L_{I}W\right)\;\\ s.t.\;W\geq 0,U_{I}\geq 0,V\geq 0,\;I=1,2,\ldots ,d, \end{array}$$
where $\lambda_{I}\geq 0$ is the balance parameter that controls the Laplacian regularization. It is worth mentioning that the different values of $\lambda_{I}$ represent the heterogeneity of multi-view data. If $\lambda_{I}=0$, GMvNMF will be simplified to MvNMF. In other words, MvNMF is a special case of GMvNMF. Therefore, the following section only shows the optimization algorithm of GMvNMF.

#### 2.3.2. Optimization of GMvNMF

The Equation (9) can be rewritten as:(10)$$\begin{array}{l} \min\sum_{I=1}^{d}\mathrm{tr}\left({\left(X_{I}-WU_{I}V\right)}^{T}\left(X_{I}-WU_{I}V\right)\right)+\lambda_{I}\mathrm{tr}\left(W^{T}L_{I}W\right)\;\\ =\mathrm{tr}\left(X_{I}^{T}X_{I}\right)-2\mathrm{tr}\left(X_{I}^{T}WU_{I}V\right)+\mathrm{tr}\left(V^{T}U_{I}^{T}W^{T}WU_{I}V\right)+\lambda_{I}\mathrm{tr}\left(W^{T}L_{I}W\right). \end{array}$$

The multiplicative iterative method was used to solve the optimization problem in Equation (10). Then the Lagrangian function f was constructed as follows:(11)$$\begin{array}{l} f=\mathrm{tr}\left(X_{I}^{T}X_{I}\right)-2\mathrm{tr}\left(X_{I}^{T}WU_{I}V\right)+\mathrm{tr}\left(V^{T}U_{I}^{T}W^{T}WU_{I}V\right)+\lambda_{I}\mathrm{tr}\left(W^{T}L_{I}W\right)\\ \;+\mathrm{tr}\left(\psi W^{T}\right)+\mathrm{tr}\left(\varphi_{I}U_{I}^{T}\right)+\mathrm{tr}\left(\mu V^{T}\right), \end{array}$$
where $\psi =\left[\psi_{il}\right]$, $\varphi_{I}={\left[\varphi_{la}\right]}_{I}$ and $\mu =\left[\mu_{aj}\right]$ are Lagrange multipliers that constrain W≥0, $U_{I}\geq 0$ and V≥0, respectively. i, l, a and j represent the subscripts of the elements in the matrix.

We separately derived the partial derivatives of W, $U_{I}$ and V of the Lagrangian function as follows:(12)$$\frac{\partial f}{\partial W}=-2X_{I}V^{T}U_{I}^{T}+2WU_{I}VV^{T}U_{I}^{T}+2\lambda_{I}L_{I}W+\psi ,$$
(13)$$\frac{\partial f}{\partial U_{I}}=-2W^{T}X_{I}V^{T}+2W^{T}WU_{I}VV^{T}+\varphi_{I},\;I=1,2,\ldots ,d,$$
(14)$$\frac{\partial f}{\partial V}=-2U_{I}^{T}W^{T}X_{I}+2U_{I}^{T}W^{T}WU_{I}V+\mu .$$

It is well known that Karush-Kuhn-Tucher (KKT) conditions [15] can be applied to solve an optimization problem with inequality constraints. By using the KKT conditions ψW=0, $\varphi_{I}U_{I}=0$ and μV=0, we can get the following update rules:(15)$$W_{il}\leftarrow W_{il}\frac{\sum_{I=1}^{d}{\left(X_{I}V^{T}U_{I}^{T}\right)}_{il}}{\sum_{I=1}^{d}{\left(WU_{I}VV^{T}U_{I}^{T}+\lambda_{I}L_{I}W\right)}_{il}},$$
(16)$${\left(U_{I}\right)}_{la}\leftarrow {\left(U_{I}\right)}_{la}\frac{{\left(W^{T}X_{I}V^{T}\right)}_{la}}{{\left(W^{T}WU_{I}VV^{T}\right)}_{la}},\;I=1,2,\ldots ,d,$$
(17)$$V_{aj}\leftarrow V_{aj}\frac{\sum_{I=1}^{d}{\left(U_{I}^{T}W^{T}X_{I}\right)}_{aj}}{\sum_{I=1}^{d}{\left(U_{I}^{T}W^{T}WU_{I}V\right)}_{aj}}.$$

Finally, we summarize the iterative process of the proposed GMvNMF model in Algorithm 1.

**Algorithm 1:** GMvNMFData Input: (XI)n×mIParameters: λIOutput: W, UI and VInitialization: W≥0, UI≥0 and V≥0Set r=1Repeat Update W by (15); Update UI by (16); Update V by (17); r=r+1;Until convergence

## 3. Results

We performed experiments with multi-view clustering and selection of co-differentially expressed genes to verify the effectiveness of the proposed method. In addition, we used jNMF [2], iNMF [8] and iONMF [9] as comparison methods. Detailed information on the experimental settings and results are shown in the following section.

### 3.1. Datasets

The Cancer Genome Atlas (TCGA) program intends to analyze the genomic variation map of cancer by using high-throughput sequencing technology [16]. As the largest cancer genome database, TCGA contains a lot of valuable and incredible information. An in-depth study of this information can help us understand, prevent, and treat cancer. In this paper, we used four multi-view datasets to analyze the performance of the proposed method. These datasets included pancreatic adenocarcinoma (PAAD), esophageal carcinoma (ESCA), colon adenocarcinoma (COAD), and head and neck squamous cell carcinoma (HNSC). Each cancer dataset contained three different types of data, such as GE, CNV, and ME. All of the above data can be downloaded from the TCGA (https://tcgadata.nci.nih.gov/tcga/). In the experiment, we performed preprocessing on the data. First, principal component analysis (PCA) was used to reduce dimensionality and remove redundant information and noise on the data. Then, the data matrix was normalized such that each row of the matrix was distributed between 0 and 1. More descriptions of multi-view datasets are summarized in Table 1.

### 3.2. Parameter Setting

In the MvNMF and GMvNMF methods, we needed to choose parameters such as k, r and $\lambda_{I}$. The values of k and r determined the size of the shared basis matrix, the subspace transformation matrix and the shared coefficient matrix. Thus, choosing a reasonable parameter value will promote the experimental results. Since the value of k means the degree of dimensionality reduction of data, it had a significant impact on the experiment. r is the rank of the matrix. If the value of r is more appropriate, then a better genetic selection result will be obtained. In other words, MvNMF and GMvNMF were sensitive to the choice of k and r. The graph regularization parameters $\lambda_{I}$ controlled the extent to which the internal geometric structure of the original data was preserved. In addition, $\lambda_{I}$ reflected the heterogeneity of data from different sources in multi-view data.

In the experiment, we empirically set $\lambda_{I}$ corresponding to different views in a multi-view data to the same value [17]. For convenience, we used the grid search algorithm to select the optimal value of the parameter. When k, r, and $\lambda_{I}$ were selected in the interval [1, 50], [1, k − 1], and [1, 100,000], respectively, MvNMF and GMvNMF achieved the best performance. The specific conditions of the selected parameters can be seen from the following figures. It should be noted that, as we can see from Figure 1a, when k=2, MvNMF had a higher accuracy in the HNSC dataset. That is to say, the value of r in MvNMF can only be 1 on the HNSC dataset. Therefore, it is not shown in Figure 1c. In summary, we can select the optimal parameters through Figure 1 and Figure 2.

### 3.3. Convergence and Computational Time Analysis

We iterated the updated rules of MvNMF and GMvNMF to approximate the local optimal solution of the objective function. In Figure 3, the convergence curves for the five methods are given (to save space, only the convergence curves on the ESCA dataset are shown). These five methods consisted of jNMF, iNMF, iONMF, MvNMF, and GMvNMF. It can be observed from Figure 3 that these five methods converged in 100 iterations. The error value is the loss function value. Additionally, the convergence criterion was that the error value tended to zero. Since MvNMF and GMvNMF had smaller error values than other methods, the convergence of MvNMF and GMvNMF is better.

In addition, we compared the execution time of these five algorithms. Experiments were executed on a PC with 3.50 GHz Intel(R) (Santa Clara, CA, USA) Xeon(R) CPU and 16G RAM. In the experiment, each method was repeated 10 times. The mean and variance were calculated. The statistics of the computational time are listed in Table 2. As can be seen from Table 2, all five methods had satisfactory running times. Because iNMF takes into account the heterogeneous effect, its computational time was affected. IONMF imposed orthogonal constraints, thus, it had the longest running time. MvNMF and GMvNMF had lower running times. This is because the experimental results showed that the decomposed matrix of our proposed method had better sparsity and lower rank than the matrix after jNMF decomposition.

### 3.4. Clustering Results

We performed clustering experiments on four multi-view datasets to verify the effectiveness of the proposed method. Multi-view clustering was performed on the shared basis matrix using the *K*-means algorithm.

#### 3.4.1. Evaluation Metrics

In order to strictly analyze the performance of multi-view clustering, we adopted multiple measures, including accuracy (AC), recall, precision, and F-measure [18,19]. The AC is defined as follows:(18)$$AC=\frac{\sum_{i=1}^{n}\delta \left(s_{i},map\left(r_{i}\right)\right)}{n},$$
where n is the number of samples contained in the dataset, $r_{i}$ is the clustering label obtained using the clustering algorithm, and $s_{i}$ is the real data label. $map\left(r_{i}\right)$ is a permutation mapping function that maps clustering labels to real data labels. The real clusters refer to the known sample labels. In addition, δ(x,y) is a delta function.

Recall, precision, and *F*-measure are a set of metrics that are widely used in clustering applications. Recall can also be called sensitivity. True-positive (TP) indicates that two data points of the same cluster are divided into the same cluster. True-negative (TN) means that two data points of the same cluster are divided into different clusters. False-positive (FP) indicates that data points of two different clusters are divided into the same cluster. False-negative (FN) refers to two data points in different clusters divided into different clusters. Recall, precision and *F*-measure are defined as follows:(19)$$recall=\frac{TP}{TP+FN},$$
(20)$$precision=\frac{TP}{TP+FP},$$
(21)$$F-measure=\frac{2}{1/recall+1/precision},$$
where *F*-measure is a comprehensive evaluation indicator that takes into account recall and precision.

#### 3.4.2. Multi-View Clustering Results

In the experiment, each algorithm is executed 50 times to reduce the impact of random initialization on multi-view clustering results. The mean and variance of performance on each multi-view data are recorded in Table 3.

According to Table 3, we can draw the following conclusions:The clustering performance of jNMF on PAAD and COAD datasets was better than iNMF, iONMF, and MvNMF. This demonstrates that improvements to the traditional NMF integration model may result in the loss of useful information, which in turn affected the clustering results. However, in the ESCA and HNSC datasets, MvNMF outperformed jNMF, iNMF, and iONMF from the overall perspective of the evaluation metrics. This shows the validity of our proposed MvNMF model, which better preserved the complementary information between multiple views.From Table 3, we can see that the precision of the GMvNMF method and the precision of the MvNMF method were similar in the four multi-view datasets. However, the GMvNMF method was at least about 18, 32 and 20% higher than the MvNMF method in terms of AC, recall, and F-measure. Therefore, the GMvNMF method had better clustering performance. This shows that it is necessary to consider the manifold structure that exists in multi-view data.Taking the four multi-view datasets in Table 3 as a whole, the proposed GMvNMF method had the best clustering performance. GMvNMF outperformed other methods by about 23, 39, 0.67, and 25%, with respect to the average values of the metrics AC, recall, precision, and F-measure. Therefore, GMvNMF is an effective integration model that takes into account the latent group structure and intrinsic geometric information between multi-view data.

### 3.5. Gene Selection Results

#### 3.5.1. Co-Differentially Expressed Gene Selection Results

It is well known that genomic alterations and genetic mutations can cause cancer [20,21]. Therefore, research on cancer genomic data is an urgent task. In the feature selection experiment, we used genomic data including GE, CNV, and ME to verify the effectiveness of the proposed method. The co-differential genes were selected on a shared coefficient matrix. Since the differential genes we selected are genes expressed in GE, CNV, and ME, the selected co-differential genes have more important biological significance.

In the experiment, we scored all the genes. These genes were then ranked in descending order of score. The higher the score of a gene, the greater its significance. We chose such a gene as a differentially expressed gene. In practice, we selected the top 500 genes of each method as co-differentially expressed genes for comparison. Then, the selected genes were placed in the GeneGards (http://www.genecards.org/) for analysis. GeneCards is a comprehensive database of human genes that provides a variety of valuable information for studying genes [22]. Table 4 lists the results of five methods for selecting co-differential genes.

In Table 4, *N* is obtained by matching the co-differential genes selected by each method to the virulence gene pool of PAAD, ESCA, COAD, and HNSC. A larger *N* indicates a higher accuracy in identifying co-differentially expressed genes. HRS represents the highest relevant score, and ARS represents the average relevant score. Relevant scores represent the degree to which a gene is associated with a disease. A higher relevant score for a gene means that the gene is likely to be a pathogenic gene. Although the ARS of MvNMF was slightly higher than GMvNMF in COAD and HNSC, the performance of co-differentially expressed genes selected by GMvNMF was better on the whole. This indicates that our method was reasonable. In order to retain the geometric structure in the data, it was necessary to add graph regularization to the method.

#### 3.5.2. Discussion of Co-Differential Genes

Table 5 lists the relevant information of the top 10 co-differential genes selected by the GMvNMF method (to save space, we only listed the top 10 genes selected in the COAD dataset.). From Table 5, we can see that *BRCA1* has the highest relevance score. *BRCA1* is a protein-coding gene involved in DNA repair. When it is mutated, the tumor suppressor protein does not form normally. This leads to the emergence of cancer. *BRCA1* has been confirmed to be related to COAD [23]. *BRCA2* is a tumor suppressor gene, which is mainly involved in the repair of DNA damage and regulation of transcription. There is literature that *BRCA2* is related to COAD [24]. As we all know, mutations in *BRCA1* and *BRCA2* increase the risk of breast or ovarian cancer [25]. Therefore, mutations in one gene may be related to the production of multiple cancers. This suggests that biologists can further study the link between COAD and breast or ovarian cancer. The protein encoded by the epidermal growth factor receptor (*EGFR*) is a transmembrane glycoprotein that is a member of the protein kinase superfamily. In addition, mutation or overexpression of *EGFR* generally triggers COAD [26].

Table 6 lists the co-differentially expressed genes with the highest relevance score selected by GMvNMF on the multi-view dataset of PAAD, ESCA and HNSC. These co-differential genes were highly likely to cause cancer. The relevance score of *EGFR* is 168.23 in the HNSC. *EGFR* is a protein-coding gene. Among its related pathways are extracellular regulated protein kinases (ERK) signaling and GE. Moreover, the importance of *EGFR* in HNSC has been widely recognized [27,28]. *EGFR* also appears in Table 5, which indicates that *EGFR* has to do with the occurrence of COAD and HNSC. This provides a new way for biologists to study COAD and HNSC.

## 4. Conclusions

In this paper, we proposed a new integrated NMF model called MvNMF for multi-view clustering and selection of co-differentially expressed genes. Considering the low-dimensional manifold structure existing in the high-dimensional multi-view sample space, the graph regularization constraint was added to the objective function of MvNMF. This new model is called GMvNMF. It effectively encodes the geometric information inherent in the data. Numerous experiments on cancer genomic data showed that our proposed GMvNMF method is more effective.

For future work, we continue to improve the model to enhance its robustness and sparsity.

## Figures and Tables

**Figure 1 genes-09-00586-f001:**
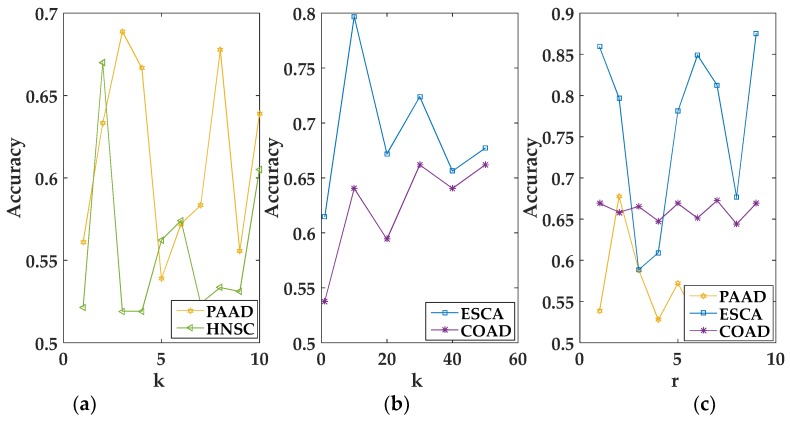
Performance of the Multi-view Non-negative Matrix Factorization (MvNMF) set with different values of k and r. (**a**) is the clustering performance of MvNMF on PAAD and HNSC about k; (**b**) is the clustering performance of MvNMF on ESCA and COAD about k; (**c**) is the clustering performance of MvNMF on PAAD, ESCA and COAD about r.

**Figure 2 genes-09-00586-f002:**
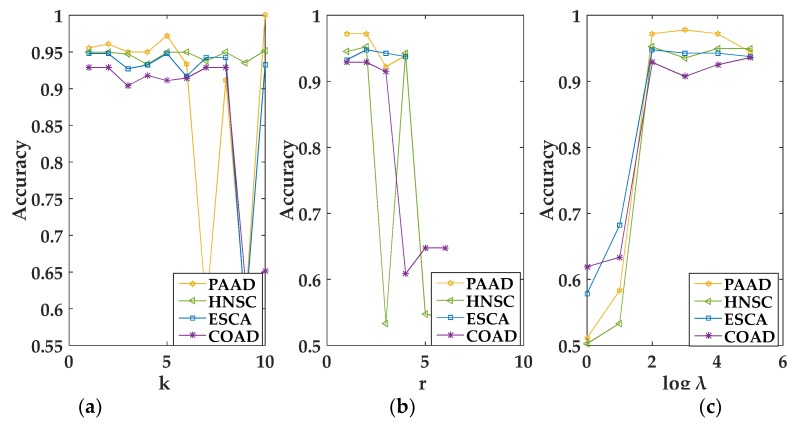
Performance of the graph regularized MvNMF (GMvNMF) set with different values of k, r and λ. (**a**) is the clustering performance of GMvNMF on PAAD, HNSC, ESCA and COAD about k; (**b**) is the clustering performance of GMvNMF on PAAD, HNSC, ESCA and COAD about r; (**c**) is the clustering performance of GMvNMF on PAAD, HNSC, ESCA and COAD about λ.

**Figure 3 genes-09-00586-f003:**
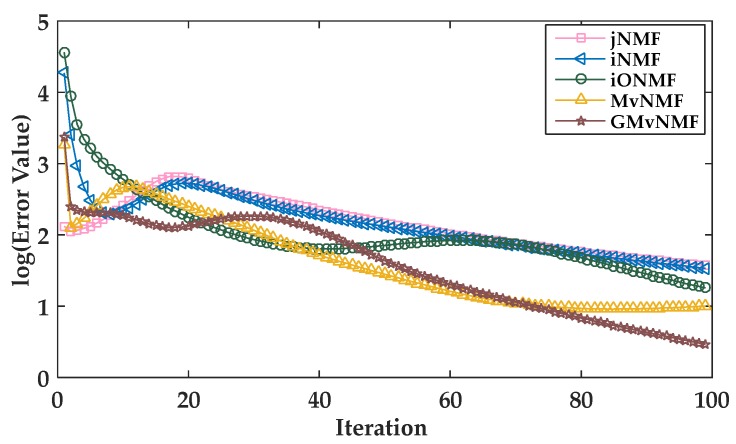
Convergence curves of joint Non-negative Matrix Factorization (jNMF), integrated NMF (iNMF), integrative orthogonality-regularized NMF (iONMF), MvNMF, and GMvNMF.

**Table 1 genes-09-00586-t001:** Description of four multi-view datasets.

Datasets	Data Types	Normal Samples	Tumor Samples	Genes
PAAD	GE, CNV, ME	176	4	19,877
ESCA	GE, CNV, ME	183	9	19,877
HNSC	GE, CNV, ME	398	20	19,877
COAD	GE, CNV, ME	262	19	16,977

Note: Datasets represent different multi-view data. PAAD: pancreatic adenocarcinoma; ESCA: esophageal carcinoma; HNSC: head and neck squamous cell carcinoma; COAD: colon adenocarcinoma; GE: gene expression; CNV: copy number variation; ME: methylation.

**Table 2 genes-09-00586-t002:** Computational time on ESCA.

Methods	Times (s)
jNMF	2.8808 ± 1.7 × 10^−4^
iNMF	3.4647 ± 1.3 × 10^−3^
iONMF	5.7375 ± 2.8 × 10^−3^
MvNMF	1.3495 ± 7.0 × 10^−5^
GMvNMF	1.0767 ± 1.4 × 10^−4^

**Table 3 genes-09-00586-t003:** The clustering performance on PAAD, ESCA, COAD and HNSC.

Methods	Metrics	jNMF	iNMF	iONMF	MvNMF	GMvNMF
PAAD	AC (%)	70.39 ± 3.71	70.30 ± 3.71	65.01 ± 2.73	63.86 ± 0.78	**95.59 ± 0.05**
	Recall (%)	61.78 ± 7.34	56.49 ± 8.30	53.17 ± 5.48	56.30 ± 2.77	**91.90 ± 5.28**
	Precision (%)	97.93 ± 0.03	**98.35 ± 0.01**	97.89 ± 0.00	97.88 ± 0.03	95.99 ± 1.92
	F-measure (%)	71.99 ± 5.26	66.92 ± 7.06	65.65 ± 4.65	69.89 ± 1.92	**92.12 ± 5.03**
ESCA	AC (%)	65.32 ± 3.70	66.42 ± 3.49	57.64 ± 0.21	68.04 ± 0.70	**93.23 ± 0.21**
	Recall (%)	51.48 ± 6.67	54.39 ± 6.55	51.90 ± 0.67	51.10 ± 3.75	**97.21 ± 0.74**
	Precision (%)	88.16 ± 5.84	88.29 ± 6.21	94.70 ± 0.20	93.51 ± 0.51	**95.20 ± 0.00**
	F-measure (%)	62.81 ± 6.60	65.61 ± 6.25	67.16 ± 0.55	64.47 ± 3.39	**95.97 ± 0.37**
COAD	AC (%)	73.91 ± 1.84	71.00 ± 1.33	66.99 ± 0.68	65.13 ± 0.03	**92.01 ± 0.01**
	Recall (%)	57.15 ± 6.54	51.28 ± 5.16	50.24 ± 2.95	47.15 ± 1.58	**94.70 ± 3.61**
	Precision (%)	90.02 ± 3.29	87.60 ± 4.52	90.18 ± 1.88	89.94 ± 0.64	**93.42 ± 0.00**
	F-measure (%)	68.25 ± 5.34	63.53 ± 5.11	63.79 ± 2.8	61.45 ± 1.58	**92.22 ± 3.23**
HNSC	AC (%)	66.75 ± 0.00	66.16 ± 0.01	66.39 ± 0.00	67.70 ± 0.03	**86.18 ± 2.67**
	Recall (%)	53.62 ± 2.19	51.18 ± 2.21	50.68 ± 2.20	55.23 ± 2.44	**87.07 ± 5.57**
	Precision (%)	95.22 ± 0.38	94.30 ± 0.39	94.03 ± 0.39	**95.53 ± 0.33**	94.93 ± 0.05
	F-measure (%)	67.85 ± 2.01	65.61 ± 1.96	65.09 ± 2.03	69.10 ± 2.20	**88.63 ± 3.30**

Note: The best experimental results are highlighted in bold.

**Table 4 genes-09-00586-t004:** Co-differential genes selection results on four multi-view datasets.

Methods	PAAD			ESCA			COAD			HNSC		
	*N*	HRS	ARS	*N*	HRS	ARS	*N*	HRS	ARS	*N*	HRS	ARS
jNMF	374	84.93	4.89	168	**76.15**	5.19	142	103.7	7.02	175	**168.23**	17.75
iNMF	375	84.93	4.84	171	**76.15**	5.31	144	103.7	7.71	175	102.98	16.65
iONMF	375	**100.56**	5.19	170	**76.15**	5.36	141	165.65	8.64	175	**168.23**	17.52
MvNMF	365	**100.56**	5.23	170	**76.15**	5.52	145	165.65	**8.66**	**177**	**168.23**	**18.00**
GMvNMF	**376**	**100.56**	**5.53**	**182**	**76.15**	**5.69**	**152**	**173.12**	8.37	**177**	**168.23**	17.60

Note: *N* is obtained by matching the co-differential genes selected by each method to the virulence gene pool of PAAD, ESCA, COAD, and HNSC. HRS represents the highest relevant score, and ARS represents the average relevant score. The best experimental results are highlighted in bold.

**Table 5 genes-09-00586-t005:** Summary of the co-differential genes selected by the GMvNMF method.

Gene ID	Gene ED	Related Go Annotations	Related Diseases	Relevance Score
672	*BRCA1*	RNA binding and ligase activity	Breast-Ovarian Cancer, Familial 1 and Pancreatic Cancer 4	173.12
675	*BRCA2*	protease binding and histone acetyltransferase activity	Fanconi Anemia, Complementation Group D1 and Breast Cancer	135.87
1956	*EGFR*	identical protein binding and protein kinase activity	Inflammatory Skin and Bowel Disease, Neonatal, 2 and Lung Cancer	104.16
3569	*IL6*	signaling receptor binding and growth factor activity	Kaposi Sarcoma and Rheumatoid Arthritis, Systemic Juvenile	58.74
4318	*MMP9*	identical protein binding and metalloendopeptidase activity	Metaphyseal Anadysplasia 2 and Metaphyseal Anadysplasia	45.57
1495	*CTNNA1*	actin filament binding	Macular Dystrophy, Patterned, 2 and Butterfly-Shaped Pigment Dystrophy	41.99
1950	*EGF*	calcium ion binding and epidermal growth factor receptor binding	Hypomagnesemia 4, Renal and Familial Primary Hypomagnesemia with Normocalciuria and Normocalcemia	40.84
5594	*MAPK1*	transferase activity, transferring phosphorus-containing groups and protein tyrosine kinase activity	Chromosome 22Q11.2 Deletion Syndrome, Distal and Pertussis	39.23
2475	*MTOR*	transferase activity, transferring phosphorus-containing groups and protein serine/threonine kinase activity	Focal Cortical Dysplasia, Type II and Smith-Kingsmore Syndrome	34.07
887	*CCKBR*	G-protein coupled receptor activity and 1-phosphatidylinositol-3-kinase regulator activity	Panic Disorder and Anxiety	23.43

Note: Gene ID represents the number of the gene. Gene ED represents the gene name.

**Table 6 genes-09-00586-t006:** Summary of the co-differential genes selected on PAAD, ESCA and HNSC.

Gene ID	Gene ED	Related Go Annotations	Related Diseases	Paralog Gene
999	*CDH1*	calcium ion binding and protein phosphatase binding	Gastric Cancer, Hereditary Diffuse and Blepharocheilodontic Syndrome 1	*CDH3*
1499	*CTNNB1*	DNA binding transcription factor activity and binding	Mental Retardation, Autosomal Dominant 19 and Pilomatrixoma	*JUP*
1956	*EGFR*	identical protein binding and protein kinase activity	Inflammatory Skin and Bowel Disease, Neonatal, 2 and Lung Cancer	*ERBB4*

Note: Paralog gene produced via gene duplication within a genome.

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
