# Peer review of "Co-differential Gene Selection and Clustering Based on Graph Regularized Multi-View NMF in Cancer Genomic Data"

_genes, 2018, doi:10.3390/genes9120586_

Round 1
Reviewer 1 Report
The authors first proposed a method named Multi-view Non-negative Matrix Factorization (MvNMF) to perform the selection of common differential genes and multi-view clustering. In addition, the authors incorporate the graph regularization constraint to MvNMF to produce graph regularized MvNMF (GMvNMF). However, the paper in general is not well written. I have had a hard time in reading this paper. My specific comments are as follows:
1. There are many grammatical issues in the manuscript. The authors should get editing help from someone with full professional proficiency in English.
2. What do the notations Wil and Hlj mean? The authors need to define the notations.
3. Could the authors provide some explanations why we need graph regularization for the gene selection?
4. What does V represent in equation (7)?
5. The trace should be introduced under (7) instead of (9)?
6. Are the k’s in Wn×k and the equation (6) the same? If not, how to determine k in (6) and (9) for the trace part?
7. The proposed methods are very sensitive to the choices of r and k. For example, in Figure 2a, as k increases, the accuracy dramatically decreases and then increases. Could the authors explain why this happened.
8. Is it possible that the authors can provide some explanations why the proposed meth- ods need less computational time?
9. How was the error value calculated in Figure 3?
Reviewer 2 Report
The authors have proposed a new Multi-view Non-negative Matrix Factorization model with applications to cancer genomic data. This is a very important and interesting topic that deserves attention though some suggestions could be given to improve the paper. I have the following comments.
1. Page 2, Line 19 (from the top of the page). Remove space between the first V and the full stop in “… matrix V .”. Add space between space between the second V and (. The d is not explained here.
2. Page, Line 1 (from the bottom of the page). What are n, k, m_I?
3. Page 3, formulas (2) and (3). What are the indices i, l and j?
4. Page 3, the paragraph after formula (3). Move citation [8] after “Yang et al.”
5. Page 3, formula (4). What is V_I?
6. Page 3, Line 4 before formula (6). Should it be “k-nearest …” instead of “K-nearest” since k (lower case) is used in formula (6)?
7. Page 3, Line 2 before formula (6). Remove space between j and the full stop. Replace “intimacy” by “proximity”.
8. Page 3, formula (6). The authors should be consistent with the notations: either E_{ij} (without comma) or E_{i,j} (with comma). See the previous lines where E_{ij} is used.
9. Page 4, Line 1 from the top of the page. It should be N_k(x_i) not N_k(x_j).
10. Page 4, formula (7). What are s_i and s_j (lower case)? Where is V in the first line of formula (7) – it is only in min_V? What is V? Is E a matrix whose elements are E_{i,j}? Does tr mean trace?
11. Page 4, Line 2 above formula (8). What is U_I? It is not explained here. Add space between V and (.
12. Page 4, he paragraph after formula (8). The n and m are not explained.
13. Page 5, the line after formula (11). Explain the indices i, l, and a.
14. Page 5, the line after formula (14). Explain KKT, which is Karush–Kuhn–Tucker.
15. Page 5, the last line. What are possible convergence criteria?
16. Page 7. Figure 2 is not commented in the text.
17. Page 8, formulas (19) and (20) and the paragraph before them. The authors should also mention that these measures are called (or related to) sensitivity and specificity as well as standard terminology such as false-positive, false-negative, true-positive and true-negative.
Round 2
Reviewer 1 Report
The authors have successfully addressed the comments in my previous report. I do not have any further comments.